# The Effect of pH and Storage Temperature on the Stability of Emulsions Stabilized by Rapeseed Proteins

**DOI:** 10.3390/foods10071657

**Published:** 2021-07-18

**Authors:** Karolina Östbring, María Matos, Ali Marefati, Cecilia Ahlström, Gemma Gutiérrez

**Affiliations:** 1Department of Food Technology Engineering and Nutrition, Faculty of Engineering, Lund University, P.O. Box 124, 221 00 Lund, Sweden; ali.marefati@food.lth.se (A.M.); cecilia.ahlstrom@food.lth.se (C.A.); 2Department of Chemical and Environmental Engineering, University of Oviedo, Julián Clavería 8, 33006 Oviedo, Spain; matosmaria@uniovi.es (M.M.); gutierrezgemma@uniovi.es (G.G.)

**Keywords:** rapeseed press cake, cold-pressed, emulsifying properties, zeta potential, turbiscan, emulsion stability

## Abstract

Rapeseed press cake (RPC), the by-product of rapeseed oil production, contains proteins with emulsifying properties, which can be used in food applications. Proteins from industrially produced RPC were extracted at pH 10.5 and precipitated at pH 3 (RPP3) and 6.5 (RPP6.5). Emulsions were formulated at three different pHs (pH 3, 4.5, and 6) with soy lecithin as control, and were stored for six months at either 4 °C or 30 °C. Zeta potential and droplet size distribution were analyzed prior to incubation, and emulsion stability was assessed over time by a Turbiscan instrument. Soy lecithin had significantly larger zeta potential (−49 mV to 66 mV) than rapeseed protein (−19 mV to 20 mV). Rapeseed protein stabilized emulsions with smaller droplets at pH close to neutral, whereas soy lecithin was more efficient at lower pHs. Emulsions stabilized by rapeseed protein had higher stability during storage compared to emulsions prepared by soy lecithin. Precipitation pH during the protein extraction process had a strong impact on the emulsion stability. RPP3 stabilized emulsions with higher stability in pHs close to neutral, whereas the opposite was found for RPP6.5, which stabilized more stable emulsions in acidic conditions. Rapeseed proteins recovered from cold-pressed RPC could be a suitable natural emulsifier and precipitation pH can be used to monitor the stability in emulsions with different pHs.

## 1. Introduction

Many of the foods we enjoy are in emulsion form, such as sauces, ice cream, and beverages. Food emulsions are mixtures of at least two immiscible liquids (e.g., water and oil), where one is dispersed as droplets in the other. The droplets are formed under mechanical shear and are stabilized by emulsifiers; these include low molecular surfactants or macromolecules, such as carbohydrates or proteins [1]. Proteins adsorb to the oil–water interface where they facilitate further droplet disruption by reducing the surface tension, while at the same time provide a protective coating that prevents the droplets from coalescing [2]. The efficacy of a protein as an emulsifier is therefore related both to the ability to generate repulsive interactions, such as steric hindering and electrostatic repulsion between droplets, as well as the ability to form a strong viscoelastic gel [3].

Rapeseed (*Brassica napus*) is an economically important oilseed crop primarily grown for the oil content and favorable fatty acid composition, leaving a protein-rich meal as an underutilized by-product [4]. Rapeseeds contain two types of storage proteins, which together constitute up to 60–80% of the protein content [5,6]. Cruciferin is a salt-soluble globulin with a molecular mass of ~300 kDa and an isoelectric point (PI) of ~7.2, and napin is a water-soluble albumin with a low molecular mass (12.5–14.5 kDa), and high PI of ~10–11.

Triacylglycerols in plants are stored as oil bodies inside the oilseed, to allow the plant to access energy rapidly when the environment is suitable for growth. Oil bodies are surrounded by phospholipids and stabilized by oil-body proteins called oleosins [7]. By this arrangement, the oil bodies are stable against aggregation and coalescence, even when subjected to severe moisture and temperature fluctuations. The stability of the oil bodies has been attributed to the oleosin protein structure [8]. The N-terminal and C-terminal domains in the oleosin protein have a scattered distribution of hydrophilic and hydrophobic regions where the hydrophobic regions are facing the lipid phase and the hydrophilic regions are facing the aqueous phase. Positively charged residues face the interior of the oil body and negatively charged residues face the exterior [9]. Oleosin also has a long double-bonded hydrophobic region, and the folding results in a 180 °C turn anchoring the protein into the oil phase. This hydrophobic region is the longest found in proteins from organisms. Oleosin (8–20% of the protein content) has a molecular mass of 15–26 kDa and an isoelectric point around ~6.5 [10,11]. Other oil-body proteins in rapeseed are caleosins with molecular weights of 25–35 kDa, which are relatively minor protein constituents associated with oleosin [12].

There are conflicting results in the literature regarding the emulsifying capacity and stability of the proteins in rapeseed. Some authors reported cruciferin to have a superior emulsifying activity index (EAI) with the formation of smaller oil droplets with higher stability compared to napin [13,14] while other studies report napin to be more efficient as an emulsifier [15,16]. Napin is a smaller protein (12.5–14.5 kDa) and was therefore suggested to have a higher diffusion rate than cruciferin, leading to higher emulsifying activity index values (EAI) in dilute emulsion systems. Ntone et al. investigated a mix of napin and cruciferin (in a mass ratio of 1:1) and attributed napins’ high emulsifying capacity to both the small size as well as its unique Janus-like structure, as 45% of the amino acids are hydrophobic and primarily located on one side of the protein. Cruciferin, with a bigger size and a more homogenous distribution of the hydrophobic domains, could not reach the interface, but appears to just weakly interact with the adsorbed layer of napins [15]. The conflicting conclusions can be attributed to different extraction methods and emulsion formulations, such as varying ionic strength and pH.

Several studies have investigated the effect of pH in rapeseed protein-stabilized emulsion. When the droplets in the emulsion are in close proximity (e.g., pH close to the isoelectric point or high ionic strength) the proteins can re-arrange themselves, which can promote droplet flocculation through increased hydrophobic attraction and disulfide bond formation between proteins adsorbed onto different droplets. On the other hand, when pH is far from the iso-electric point or when the formulation has low ionic strength, interactions at the interface may lead to a stronger interfacial membrane, which provides better protection against droplet coalescence [3]. Chang et al. found the emulsion droplet size to be smaller at pH 3 compared to pH 5 and 7 in emulsions stabilized by rapeseed protein isolated by the salt method [17]. Tan et al. found higher emulsifying capacity and stability of a rapeseed isolate in pH 9 compared to pH 4, which was confirmed by Pirestani et al. [14,18]. The differences are probably due to the different isoelectric points of the protein isolates used. There are also a few studies investigating emulsion stability over time. Alashi et al. found rapeseed protein-stabilized emulsions to have smaller droplet sizes at higher pH 9 compared with pH 4 both directly after homogenization and after 7 days incubation in 4 °C and 25 °C [19]. Pirestani et al. found the droplet size to increase from pH 4 to 5 followed by a decrease in pH 7 after 14 days of incubation when using a protein isolate with an isoelectric point of pH 5 [18]. The emulsifying capacity as an effect of pH seems to be dependent on the extraction technique, which governs the ratio between the different types of rapeseed proteins and the isoelectric point, but also the emulsion pH seems to affect the stability during storage.

Although the emulsion capacity and stability of rapeseed protein-stabilized emulsions have been investigated using various pH and processing methods, the effects of storage duration, temperature, and pH, which are critical factors for the incorporation of rapeseed proteins into foods, have not been fully examined. More specific, the knowledge regarding emulsion stability over a longer period of time is lacking.

Therefore, the aim of the present work was to study the effect of precipitation pH, emulsion pH, and storage temperature on emulsions stabilized by rapeseed protein over a six month period.

## 2. Materials and Methods

### 2.1. Sources of Materials and Chemicals

Industrially cold-pressed rapeseed press-cake (*B. napus*) was a kind gift from Gunnarshögs Jordbruk AB, Hammenhög, Sweden, a producer of cold-pressed rapeseed oil. The oil temperature during extraction did not exceed 35 °C and no solvents were used during extraction. Citric acid (C_6_H_8_O_7_, CAS 77-92-9) was purchased from Merck (Darmstadt, Germany). Miglyol 812 was purchased from Sasol (Witten, Germany). De-oiled lecithin from soy was purchased from Cargill (Minneapolis, MN, USA). All other chemicals were of analytical grade.

### 2.2. Production of Rapeseed Protein Concentrates

The protein recovery procedure was modified from the procedure previously described by Wijesundera [20]. The rapeseed press cake was stored in the freezer (−18 °C) until the start of the protein recovery process. Rapeseed press cake (50 g per batch) was ground by a Grindomix GM 200 knife mill for 20 s (Retsch GmbH, Haan, Germany). The pulverized press cake was hydrated with tap water (1:10 *w*/*w*) and the pH was adjusted to 10.5 with 2M NaOH. The dispersion was mixed with a Eurostar digital stirrer equipped with a Rushton impeller for 10 min (IKA Labortechnik, Staufen, Germany) at 700 rpm, pH was re-adjusted to 10.5, and the dispersion was incubated under mixing at room temperature for 4 h. After incubation, the dispersion was centrifuged (Avanti^®^ J-15R Centrifuge, Beckman Coulter, Brea, CA, USA) for 20 min at 4700× *g* and 20 °C. The supernatant was collected, and pH was adjusted with citric acid powder to either pH 6.5 or 3.0. The dispersions were again centrifuged for 20 min at 4800× *g* and 20 °C (Beckman Coulter, Allegra^®^ X-15R Centrifuge, Brea, CA, USA) and the sediment was collected. Extractions were performed in triplicates for each precipitation pH. The subsequent sediments were freeze-dried using a laboratory freeze dryer (Hetosicc freeze dryer CD 12, Birkerod, Denmark). Rapeseed sediments were distributed into aluminum trays to form a layer of 10 mm maximum. The samples were frozen at −18 °C for 24 h before freeze-drying. The plate temperature was 20 °C, the condenser −50 °C and the vacuum pressure of the dryer was 0.02 mbar. The residence time for the samples in the freeze-dryer was seven days. After the termination of freeze-drying, the samples were put in a desiccator for two days to remove any remaining moisture. The resulting powder was stored in the freezer (−18 °C) until evaluation. Rapeseed protein powder precipitated at pH 3 is hereafter referred to as RPP3 and protein powder precipitated at pH 6.5 is referred to as RPP6.5.

The protein concentration of the dehydrated rapeseed protein powders was analyzed by the Dumas method. Nitrogen content was determined by the elemental analyzer Flash EA 1112 (Thermo Electron Co., Waltham, MA, USA) blanked with air and with aspartic acid as reference. Approximately 25 mg of material was placed in a tin cylinder (diameter 30 mm) for analysis. A conversion factor of 6.25 was used to calculate protein content. Each powder was analyzed in triplicate.

### 2.3. Emulsion Formulation

Oil-in-water emulsions (33%) were prepared in glass test tubes with 4.67 mL continuous phase (phosphate-buffered saline: 0.01 M phosphate, 0.0027 M KCl, 0.137 M NaCl, pH 7.4), 2.33 mL dispersed phase (Miglyol 812), and 10 mg rapeseed protein/mL oil. This particular protein concentration was chosen based on a preliminary study where different protein concentrations were evaluated and concentrations above 10 mg rapeseed protein/mL oil did not yield any further reduction in droplet size (data not shown). Soy lecithin was used as control and screening was conducted prior to the onset of the experiment to scale the emulsion droplet size. It was found that 10 mg soy lecithin stabilized emulsion droplets in the same size as rapeseed protein and that this concentration was low enough to prevent over-saturation at the oil–water interface.

Rapeseed powder (RPP3 and RPP6.5) or soy lecithin was added to the phosphate-buffered saline and was allowed to rehydrate for 10 min followed by homogenization at 20,000 rpm for 60 s. Immediately after the emulsification, the pH was adjusted to 3.0, 4.5, and 6.0 respectively with citric acid or NaOH. The emulsion pH range was based on the common food emulsion application mayonnaise which had a pH of 4.5 (data not shown). We wanted to investigate how rapeseed protein stabilized emulsions with the same pH as mayonnaise, but also more acidic formulations as well as formulations closer to neutral pH. Benzoic acid (0.1%) was added to control the microbial growth and the addition did not affect the emulsifying properties. Miglyol oil was added, and the emulsions were homogenized with a high-performance rotor–stator homogenizer (Silent Crusher M, Heidolph Instruments GmbH& Co. KG, Schwabach, Germany) at 20,000 rpm for 60 s. Each formulation was prepared in triplicate.

A graphical representation of the formulation and characterization of emulsions is provided in Figure 1.

### 2.4. Particle Size Analysis

The particle size distribution of the oil-in-water (O/W) emulsions was measured by laser diffraction using a Malvern Mastersizer S (Malvern Instruments Ltd., Worcestershire, UK). The pump velocity was 2000 rpm, the obscuration was between 10–20% and the refractive index (RI) was 1.45 for the miglyol oil and 1.33 for the water. Each emulsion was measured two times and the average was reported.

### 2.5. Zeta Potential

Zeta potential was analyzed with a Malvern Zetasizer Nano ZS (Malvern Instruments Ltd., Worcestershire, UK) at 25 °C. The emulsions were diluted a hundred times in deionized water prior to analysis and duplicate measurements with three consecutive runs were performed.

### 2.6. Surface Tension

The surface tension of aqueous dispersions with different protein concentrations was measured at 25 °C. Rapeseed protein powder was added to the phosphate buffer in the same concentration as the emulsion formulation. All surface tensions were measured following the Du Noüy’s platinum ring method [21] using a KSV Sigma 700 tensiometer (KSV Instruments Ltd., Helsinki, Finland).

### 2.7. Emulsion Stability over Time in Different Storage Temperatures

Emulsions were incubated without dilution in cylindrical glass cells at either 4 °C or 30 °C for six months. Emulsion destabilization phenomena were determined by static multiple light scattering using a Turbiscan Lab Expert (Formulaction Co., L’Union, France), and transmitted and backscattered light was monitored as a function of time and sample cell height. The optical reading head scanned the sample in the cell, providing transmission (TS) and backscatttering (BS) data every 40 µm as a function of the sample height (in mm). These profiles provide a macroscopic fingerprint of the emulsions at a given time providing information regarding changes in droplet size, the appearance of creaming, or clarification processes. Thus, the height of the clarification front and migration velocity of creaming oil droplets as a function of time can be monitored [22]. The vials were scanned every day for the first week and thereafter approximately every second week for six months in total.

### 2.8. Statistical Analysis

Statistical analyses were performed on zeta potential, surface tension, and emulsion droplet size (*d*_43_) using SPSS Statistics 26 (IBM, Armonk, NY, USA). Equality of variance was tested with the Shapiro-Wilks test. Data for surface tension and emulsion droplet size were normally distributed, and a General Linear Model (GLM) with Tukey’s test was performed to investigate significant differences. Data for zeta potential were not normally distributed and therefore, a Kruskal–Wallis test with Bonferroni-adjusted pairwise comparison was performed to investigate any significant differences in the non-parametric data sets. Differences were considered significant if *p* < 0.05.

## 3. Results and Discussion

Protein was extracted and precipitated from rapeseed press cake at two different acidic pHs in order to investigate the emulsifying properties of the two rapeseed protein concentrates. The protein concentration 56.4% (dry basis) for protein precipitated at pH 3 (RPP3) and 57.4% (dry basis) for protein precipitated at pH 6.5 (RPP6.5).

### 3.1. Effect of Precipitation pH on Emulsifying Properties

#### 3.1.1. Zeta Potential

The zeta potential was determined for emulsions at different pHs stabilized by RPP3, RPP6.5, and soy lecithin (Figure 2). Emulsions stabilized by soy lecithin exhibited by far the lowest zeta potential, −49 mV to −65 mV depending on emulsion pH where the lowest zeta potential was found at the highest emulsion pH investigated. Emulsions stabilized by rapeseed protein exhibited significantly higher zeta potential compared to soy lecithin. Emulsions stabilized by RPP6.5 had significantly higher zeta potential (21–0.5 mV) compared to RPP3.5 (3.1–19.5 mV). The results showed that the net charge was zero at pH 3.5 for RPP3, and the corresponding pH for RPP6.5 was around pH 6 (Figure 2). This corresponds to the precipitation pHs used in the extraction method where different pHs were applied in the precipitation step (pH 3 and pH 6.5), and proteins with their lowest solubility at each pH were recovered.

The zeta potential values for emulsions stabilized by RPP3 are in agreement with Tang et al. who reported that protein extracted from cold-pressed rapeseed press cake had a zeta potential around −15 mV when the pH in the emulsion was neutral [23]. Tang et al. extracted canola protein with a salt-based method instead of the pH shift method used in the present paper. The protein composition in the study by Tang et al. was more similar to RPP3 in the present study since no iso-electrical focusing had been applied towards the oleosin fraction with an isoelectric point of 6.5 [20]. Chang et al. also used the salt method for the extraction of rapeseed protein and reported zeta potential values around 30–15 mV in the span of pH 3 to pH 7. A net neutral charge was found at pH 6.2 [17]. It is well known that the isoelectric point of rapeseed protein highly depends on botanical variety, which could be the reason for the deviating zeta potential in the literature.

#### 3.1.2. Emulsion Droplet Size

For all emulsion pHs investigated, RPP6.5 stabilized emulsions with significantly smaller droplet sizes (*p* < 0.05) compared with emulsions stabilized by RPP3 (Figure 3). Due to the reduced solubility for oleosin at pH 6.5 during the extraction process, it can be expected that the proteins precipitated at pH 6.5 (RPP6.5) had a higher concentration of oleosin compared with the proteins precipitated at pH 3.0 (RPP3) [20]. Oleosin has a unique structure with two amphipathic domains associated with the oil surface and a long hydrophobic domain anchoring into the oil droplets, providing stability to emulsion droplets.

The droplet size of emulsions stabilized by rapeseed protein was reduced with increased emulsion pH, with a significantly smaller emulsion droplet size at emulsion pH 6 compared to the droplet size at pH 3 and 4.5. This indicates an increased emulsifying capacity in pHs close to neutral, independent of precipitation pH in the extraction process (Figure 3). For RPP3, with a net neutral charge at pH 3.5, this can be explained by increased solubility and an increased net charge at pH’s away from the isoelectric point. Increased solubility allows the proteins to rapidly associate with the oil–water interface, and the increased net charge facilitates repulsion and hinders coalescence. However, RPP6.5, with a net neutral charge at pH around 6 (Figure 2), also stabilized smaller emulsion droplets with higher emulsion pH. This was unexpected and can be related to the higher concentration of oleosins with high emulsifying properties in this protein concentrate. Both Tan et al. and Alashi et al., reported the *d*_43_ for emulsions stabilized with rapeseed proteins to be around 80 µm in emulsions at pH 4, and 60 µm at pH 7 [14,19]. This is in line with the present study for RPP3 at the lower pH range, although RPP6.5 stabilized smaller droplets. Both authors used the pH shift method but used a higher pH (pH 12) in the extraction phase, and the proteins were precipitated at pH 4. The high alkali pH in the studies by Tan et al. and Alashi et al. can explain the larger droplet size in the reported studies since extreme alkali conditions have been demonstrated to induce denaturation with reduced protein solubility and altered emulsifying properties as a consequence [24]. Rapeseed protein isolates/concentrates recovered by the pH shift method are sometimes regarded as less efficient emulsifiers [5], but the present study suggests that the emulsifying properties depend on how the proteins are extracted: by using a mild alkali extraction pH and optimizing the precipitation pH, the emulsifying properties can be significantly improved.

Emulsions stabilized by soy lecithin exhibited the opposite trend than emulsion stabilized by rapeseed proteins, with increased droplet size at higher emulsion pHs. Emulsions formulated at pH 3 had significantly smaller droplets compared to emulsions at pH 4.5 and 6 (Figure 3). Overall, soy lecithin was more efficient as an emulsifier in acidic emulsions (droplet size of 30 µm in emulsions with pH 3), whereas RPP6.5 was more efficient in emulsions at pH 4.5 and above (Figure 3). In emulsions with pH 6, both RPP6.5 and RPP3 stabilized significantly smaller droplets compared to soy lecithin. It is evident that plant components from different crops are affected differently by emulsion pH and that some are more efficient emulsifiers in acidic environments (as soy lecithin), whereas others are more efficient in emulsion pHs close to neutral (as rapeseed protein).

#### 3.1.3. Droplet Size Distribution

The emulsions were polydisperse, with a multimodal droplet size distribution owing to protein aggregation in the continuous phase. This result is in line with Chang et al. and Pirestani et al. [17,18]. The droplet size distribution in emulsions with pH 3 and 4.5 stabilized by RPP3 showed one dominating peak around 76–89 µm representing emulsion droplets and a smaller peak around 5 µm representing protein aggregates (Figure 4A). Emulsions with pH 6 showed the same profile but displayed an additional peak around 190, which might be air bubbles. Emulsions stabilized by RPP6.5 showed a similar trend with one dominating peak accompanied with a smaller peak around 4 µm (Figure 4B), although the dominating peak represented emulsion droplets with smaller particle size (48–65 µm) compared with emulsions stabilized by RPP3 (76–89 µm). Emulsions stabilized by soy lecithin showed one dominating peak around 76 µm for emulsions with pH 6 and 4.5 and 35 µm for emulsions with pH 3, as well as an additional small peak with the size of 2–10 µm (Figure 4C). Other research works were done using soy lecithin as O/W emulsion stabilizer, and lower particle size was registered, around 1–15 µm [25]. However, it is important to point out that in that study around 10 times more protein was used and the agitation rate was considerably lower than in the present study. In other studies, 10 µm of droplet size were also reported when using other preparation techniques, such as microfluidization [26]. Moreover, it is important to point out that in the present study the droplet size distributions shown are presented in terms of volume, which could easily arise to larger values than when the droplet size distributions are reported in terms of number [1]. Smaller sizes can also be found in the literature when different types of plant proteins are combined with polysaccharides or surfactants to stabilize O/W emulsions, obtaining droplet sizes around 1–50 µm [26,27,28].

#### 3.1.4. Stability Mechanisms

An emulsion can be stabilized by three different mechanisms or a combination of electrical repulsion, decreased surface tension, or steric hindrance [3]. From the zeta potential experiments, it was evident that the main stabilization mechanisms of rapeseed proteins in the formulation used were not electrical repulsions. Although rapeseed proteins precipitated at pH 6.5 had a zeta potential around zero in emulsions with pH 4.5 and 6, the proteins could still stabilize emulsion droplets smaller than soy lecithin in this emulsion pH range. In order to investigate which stabilization mechanism was the main driver, the surface tension was analyzed. Pure buffer had a surface tension of 70–72 mN/mm. When rapeseed protein concentrate was added in the same concentration as the emulsion formula, the surface tension in phosphate buffer was reduced to 40 mN/mm for RPP3 and 36 mN/mm for RPP6.5, indicating that proteins in RPP6.5 possessed greater ability to reduce the surface tension compared with RPP3. The present study shows that RPP6.5 stabilized smaller emulsion droplets compared to RPP3, and since the zeta potential was low, and the reduction of surface tension was greater for RPP6.5, we suggest that both surface tension and steric hindrance are mechanisms involved in the stabilization of the oil–water interface.

### 3.2. Emulsion Stability during Six Months of Storage

#### 3.2.1. Emulsions with pH 6 Stored at 4 °C and 30 °C

The backscattering stability (BS) profiles (Figure 5, Figure 6, Figure 7 and Figure 8) obtained by the Turbiscan instrument, shows the stability of emulsions stored at 4 °C and 30 °C respectively, over six months of storage. Emulsions were prepared at pH 3, 4.5, and 6 and the stabilizers investigated were RPP3, RPP6.5 and soy lecithin.

For emulsions prepared at pH close to neutral (pH 6) and stored at 4 °C (Figure 5), it was observed that both rapeseed protein concentrates (RPP3 and RPP6.5) presented higher stability during the studied period, compared to emulsions formulated using soy lecithin as a stabilizer. A clarification layer was observed in the BS profiles in all cases indicating creaming. This phenomenon was exponential during the first 25 and 50 days for RPP3 and RPP6.5 respectively. After that, the increase of the clarification layer was reaching a plateau. Similar observation was also reported by Wijesundera et al. in a study where rapeseed protein-stabilized emulsions were stored at 40 °C for 40 days. They found that creaming occurred during the first ten days and that the phenomenon was more pronounced in emulsions with low pH [20]. In the case of the emulsions prepared with the precipitated proteins (Figure 5A,B), the clarification layer has a height of less than half of the cell after six months of storage. However, for the emulsion stabilized with soy lecithin (Figure 5C), the clarification layer arose to more than half of the cell after the second day of storage while the emulsion top part presented a fully destabilized system. Partial destabilization was also observed for emulsions stabilized with RPP6 after two months of storage presenting a reduction of BS at the top part of the cell, which indicates the initial appearance of coalescence of large drops, which was not the case for the emulsion stabilized by RPP3.

A similar trend was observed for emulsions prepared at pH 6 stored at 30 °C (Figure 6), but the instability observed was larger compared to storage at 4 °C. The emulsions stabilized by RPP3 presented reasonable stability during the six months of storage since only a clarification layer was observed. In the case of emulsions stabilized with RPP6, a large instability was observed after two months of storage by the presence of three different phases: a clarification layer at the bottom part of the cell, an emulsion phase in the middle, and the presence of free oil with some free protein at the top part of the cell. For emulsions formulated with soy lecithin, a similar behavior as for RPP6 was observed from the first day of the monitoring time.

Emulsions were also prepared at pH 4.5, as an intermediate condition between both pHs presented (pH 3 and 6). As a general trend, emulsions stabilized by precipitated rapeseed protein were more stable than those stabilized by soy lecithin. At 4 °C, both precipitated protein concentrates (RPP3 and RPP6.5) presented large stability. However, at 30 °C, emulsion stabilized by RPP3 protein, which started to present a large creaming phenomenon at the top part of the cell after 70 days of storage. Appendix A contain BS profiles of samples prepared at pH 4.5 and stored at 4 and 30 °C respectively.

#### 3.2.2. Emulsions with pH 3 Stored at 4 °C and 30 °C

Emulsions formulated at pH 3 with RPP6.5 as an emulsifier presented the highest stability after six months of storage at 4 °C (Figure 7). A clarification of around 10 mm was observed at the bottom part of the cell after one day, increasing with time up to 18 mm at the end of the monitoring time. Similar behavior was observed for emulsions stabilized by RPP3 at the same conditions, with a clarification layer of 15 mm after the first day. However, coalescence took place, especially after three months of storage, leading to a clear destabilization of the formulated system. Soy lecithin presented a good stability behavior in emulsions at pH 3 during the first ten days. Thereafter, a large instability was observed and the system separated into two separate phases: one clarification aqueous phase at the bottom and a layer with mainly free oil at the top part of the cell.

At 30 °C (Figure 8), all phenomena observed were accelerated within less than one day. Emulsion instability was observed for emulsions stabilized by RPP3, while emulsions stabilized by soy lecithin were not affected by the storage temperature to the same extent. However, for emulsions stabilized by RPP6.5, the stability was reduced after storage at 4 °C compared to 30 °C. Coalescence of larger oil droplets at the top part of the cell were observed after 20 days of storage, but the emulsion phase remained present during the studied period of six months.

Emulsions at pH 3 stabilized by RPP6.5 showed a slower increase in the clarification layer than emulsions stabilized by RPP3, probably due to the smaller droplet sizes observed for emulsions stabilized by RPP3 at the indicated pH (Figure 4). There were clear differences in emulsion stability during storage, depending on the emulsifier used. Even though emulsions stabilized by soy lecithin had the smallest droplet size at the onset of the experiment, these emulsions demonstrated large instability during storage. The clarification layer formed in emulsions prepared with RPP3 or RPP6.5 presented proteins in the aqueous phase compared to the clarification obtained in emulsions stabilized by soy lecithin that had a translucent character indicating that the aqueous phase was mainly water; and therefore, the soy lecithin remained mainly in the oily phase in the unstable emulsions.

#### 3.2.3. Summary of Stability as a Function of Emulsion pH and Storage Temperature

Figure 9 shows a comparative overview of the different emulsion phases obtained for each formulation prepared and studied at different pHs and stored at 4 °C and at 30 °C.

The precipitation pH in the protein extraction process, emulsion pH and storage temperature all affected the emulsion stability. Emulsions stabilized by RPP3 expressed increased stability with increased pH and emulsions with pH 6 were stable during six months with no sign of coalescence. Destabilization processes were accelerated with increased storage temperature although emulsions with pH 6 were stable also during storage at 30 °C. These findings are in line with Alashi et al. who reported emulsions stabilized by rapeseed protein precipitated at pH 4 (similar to RPP3 in the present study) to have higher stability at pH 7 than pH 4 [19]. Emulsions stabilized by RPP6.5 displayed a different behavior with higher stability in more acidic conditions, such as pH 3 and 4.5. The most stable emulsion in the present study was stabilized by RPP6.5 at pH 3 and stored at 4 °C. These emulsions were more sensitive to elevated storage temperature than those stabilized by RPP3. Both rapeseed protein concentrates stabilized the most stable emulsions in pH ranges away from their respective precipitation pH. During these conditions, the net charge is increased, which facilitated repulsion and counteracts coalescence with increased emulsion stability as a consequence [3]. Precipitated rapeseed proteins presented a larger ability to stabilize emulsions for a longer time than soy lecithin. The stability of emulsions stabilized by soy lecithin increased with decreased emulsion pH. The most favorable formulation was emulsion pH 3 and storage temperature 4 °C, where coalescence occurred. In all other conditions, a large phase of destabilized free oil was obtained in the top phase of the emulsion indicating a destabilized system, and the destabilization process was accelerated by increased storage temperature.

However, rapeseed protein presented larger stability than other types of proteins used in the literature to stabilize O/W emulsions, such as faba bean protein [29], where emulsions were stored for seven days and destabilization, such as creaming and clarification, was observed after the first day. Moreover, Tong et al. have used soybean protein to stabilize O/W emulsions analyzing emulsion stability for less than one hour, observing the coalescence phenomena and creaming, due to the migration of oil droplets to the upper part of the cell, and even in some cases the presence of free oil due to the reduction of backscattering at the top part [30]. A similar behavior was observed in the present study for samples mainly stabilized by soy lecithin and stored at 30 °C.

## 4. Conclusions

Emulsifying properties and emulsion stability were assessed for rapeseed protein precipitated at two different pHs with soy lecithin as control. Soy lecithin had a significantly larger zeta potential compared to rapeseed protein precipitated at pH 3 (RPP3) and pH 6.5 (RPP6.5). RPP6.5 stabilized smaller emulsion droplets than RPP3 independent of emulsion pH. Soy lecithin stabilized smaller droplets in acidic conditions whereas rapeseed protein stabilized smaller droplets in emulsions with a pH close to neutral. After six months of storage, it was concluded that both rapeseed protein precipitates stabilized the most stable emulsions away from the precipitation pH. Emulsions stabilized by RPP3 had higher stability with increased pH (with an optimum of pH 6); whereas emulsions stabilized by RPP6.5 had higher stability with decreased pH (optimum pH 3). Both RPP3 and RPP6.5 were able to stabilize emulsions with high stability over six months. Destabilization processes were accelerated with increased storage temperature. Emulsions stabilized by soy lecithin expressed lower stability, especially during storage at 30 °C, and free oil layers were visible in emulsions with pH at and above 4.5. Rapeseed proteins recovered from cold-pressed rapeseed press cake could be a suitable natural emulsifier and precipitation pH can be used to monitor the stability in emulsions with different pHs. The present study shows that rapeseed protein can be a useful emulsifier with excellent stability for food emulsions in the range of pH 3–6. Further studies are needed to reduce the bitter flavor and improve the dark color of the rapeseed protein concentrate.

## Figures and Tables

**Figure 1 foods-10-01657-f001:**
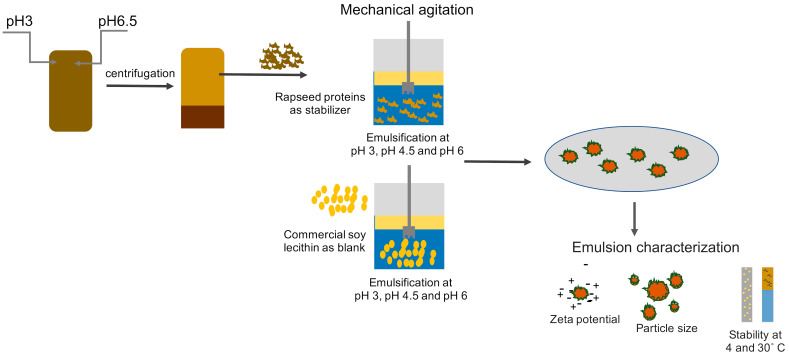
Graphical representation of the formulation and characterization of emulsions stabilized by rapeseed protein and soy lecithin.

**Figure 2 foods-10-01657-f002:**
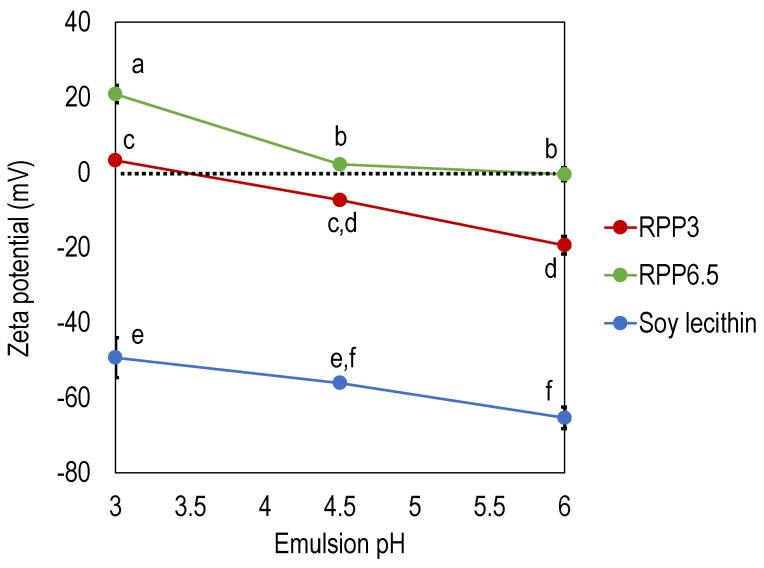
Zeta potential of emulsions with varying pH, stabilized by rapeseed protein precipitated at pH 3 (RPP3) and pH 6.5 (RPP6.5) and soy lecithin. Data is an average from four measurements for each formulation. Different letters indicate significant differences between the data points, *p* < 0.05.

**Figure 3 foods-10-01657-f003:**
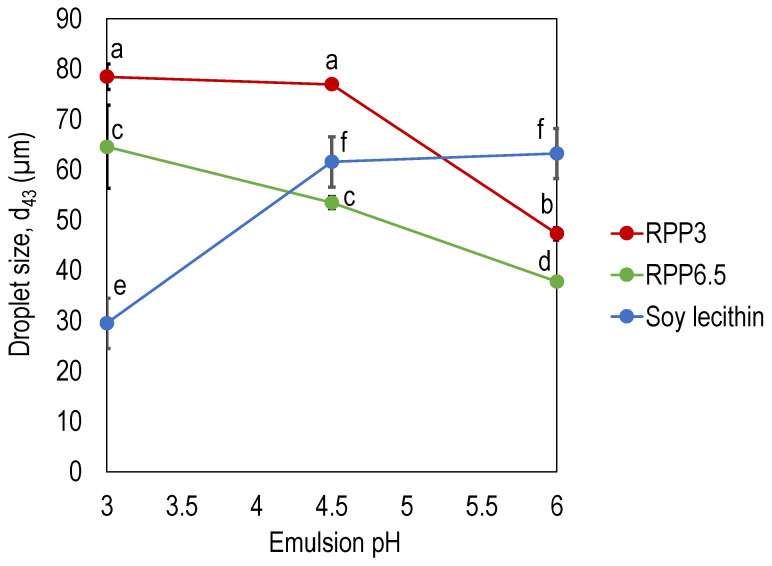
Emulsion droplet size (*d*_43_) as a function of emulsion pH in emulsions stabilized by rapeseed protein precipitated at pH 3 (RPP3) pH 6.5 (RPP6.5), and soy lecithin. Emulsions were 33% oil-in-water emulsions produced by high shear homogenization. Emulsifier concentrations in the emulsions were 10 mg protein/mL oil for the rapeseed formulations and 10 mg/mL oil for the de-oiled lecithin formulations. Data is an average from four measurements for each formulation. Different letters indicate significant differences between the different datapoints, *p* < 0.05.

**Figure 4 foods-10-01657-f004:**
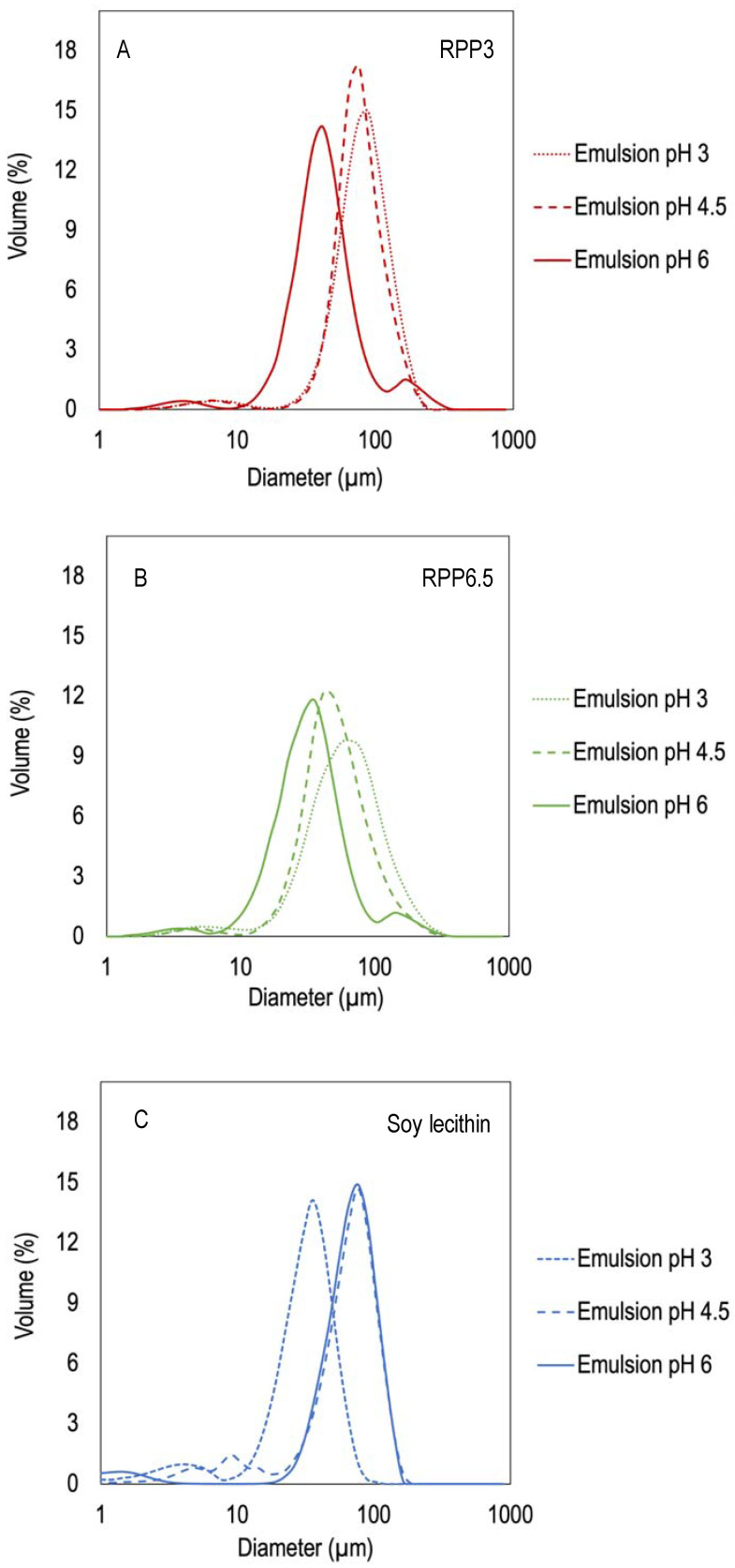
Size distributions of emulsions stabilized by rapeseed proteins precipitated at different pH and soy with varying emulsion pHs, (**A**) Emulsions with rapeseed protein precipitated at pH 3 as emulsifier (RPP3), (**B**) Emulsions with rapeseed protein precipitated at pH 6.5 as emulsifier (RPP6.5). (**C**) Soy lecithin as emulsifier, Protein concentrations in the emulsions were 10 mg protein/mL for the rapeseed formulations and 10 mg/mL oil for the de-oiled lecithin formulations.

**Figure 5 foods-10-01657-f005:**
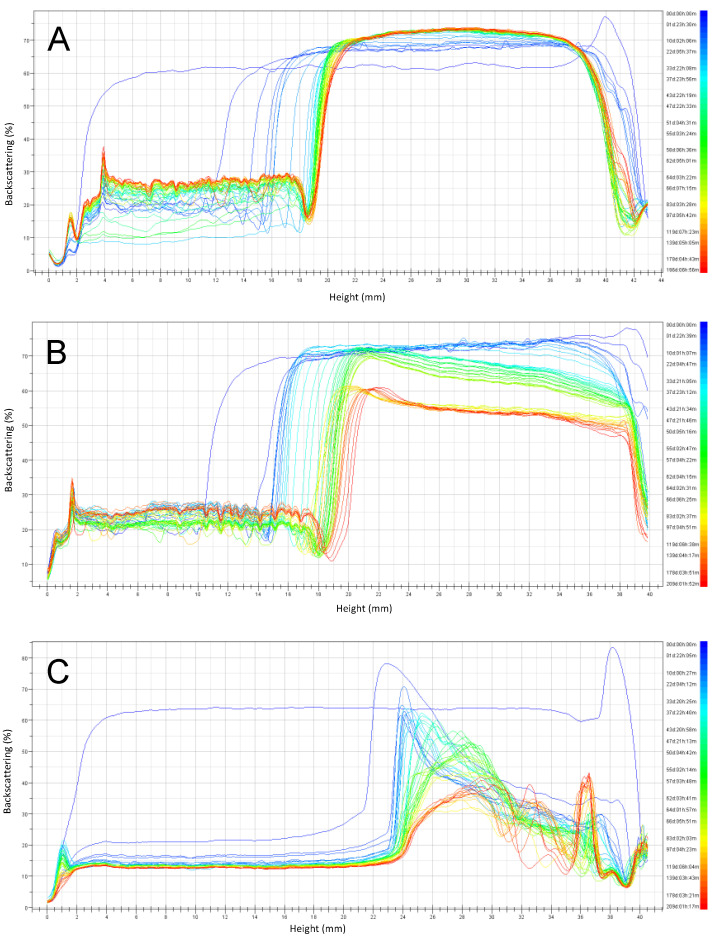
Backscattering profiles of emulsions prepared at pH 6 and stored six months at 4 °C. (**A**) Emulsions stabilized by rapeseed protein precipitated at pH 3 (RPP3), (**B**) Emulsions stabilized by rapeseed protein precipitated at pH 6.5 (RPP6.5), (**C**) Emulsions stabilized by soy lecithin.

**Figure 6 foods-10-01657-f006:**
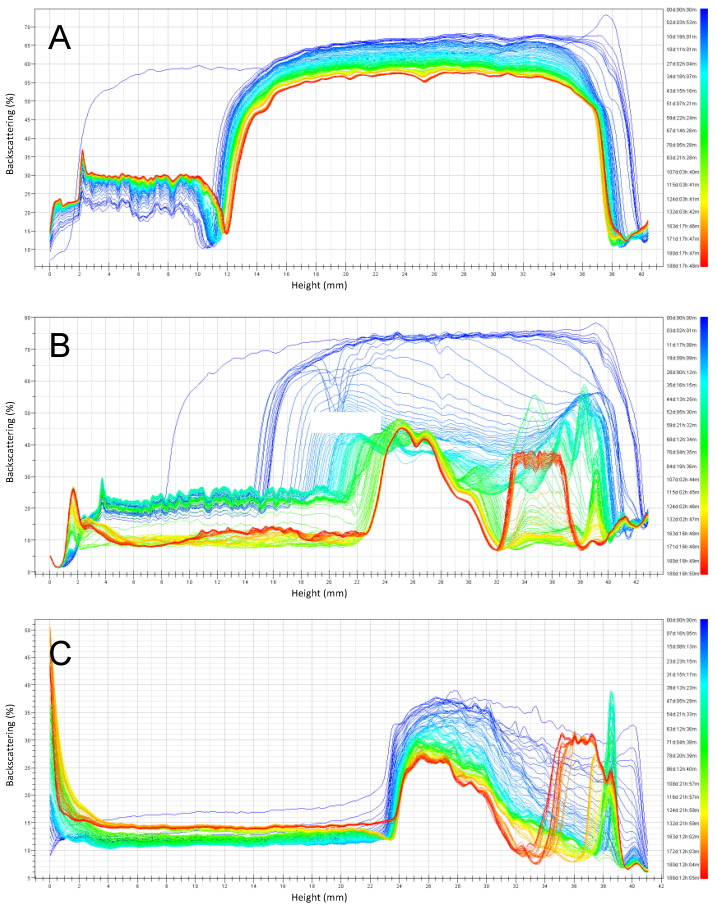
Backscattering profiles of emulsions prepared at pH 6 and stored for six months at 30 °C. (**A**) Emulsions stabilized by rapeseed protein precipitated at pH 3 (RPP3), (**B**) Emulsions stabilized by rapeseed protein precipitated at pH 6.5 (RPP6.5), (**C**) Emulsions stabilized by soy lecithin.

**Figure 7 foods-10-01657-f007:**
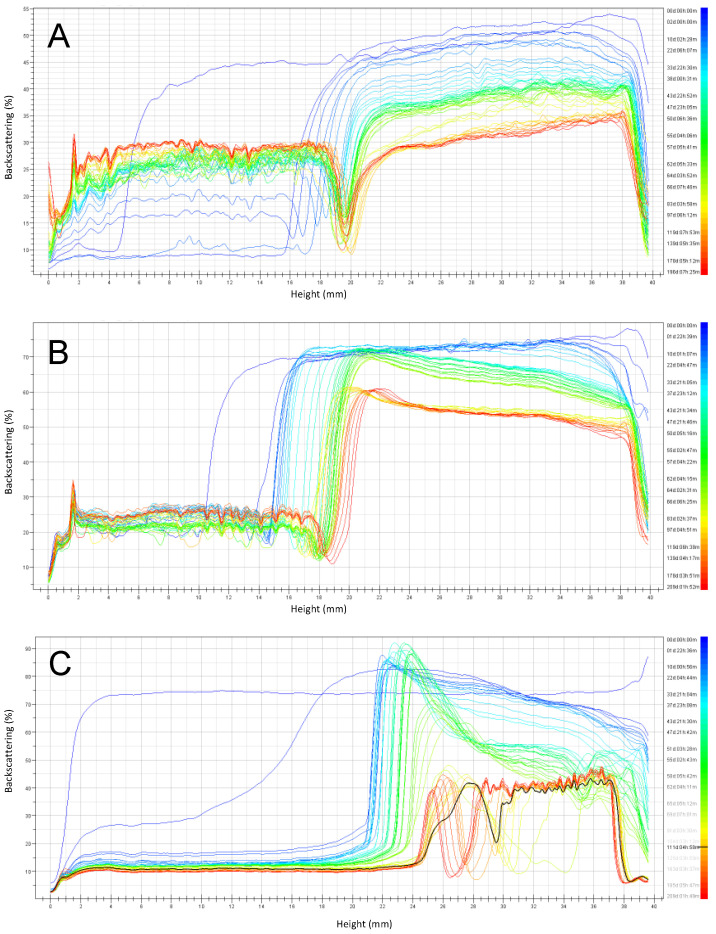
Emulsion backscattering profiles prepared at pH 3 and stored for six months at 4 °C. (**A**) Emulsions stabilized by rapeseed protein precipitated at pH 3 (RPP3), (**B**) Emulsions stabilized by rapeseed protein precipitated at pH 6.5 (RPP6.5), (**C**) Emulsions stabilized by soy lecithin.

**Figure 8 foods-10-01657-f008:**
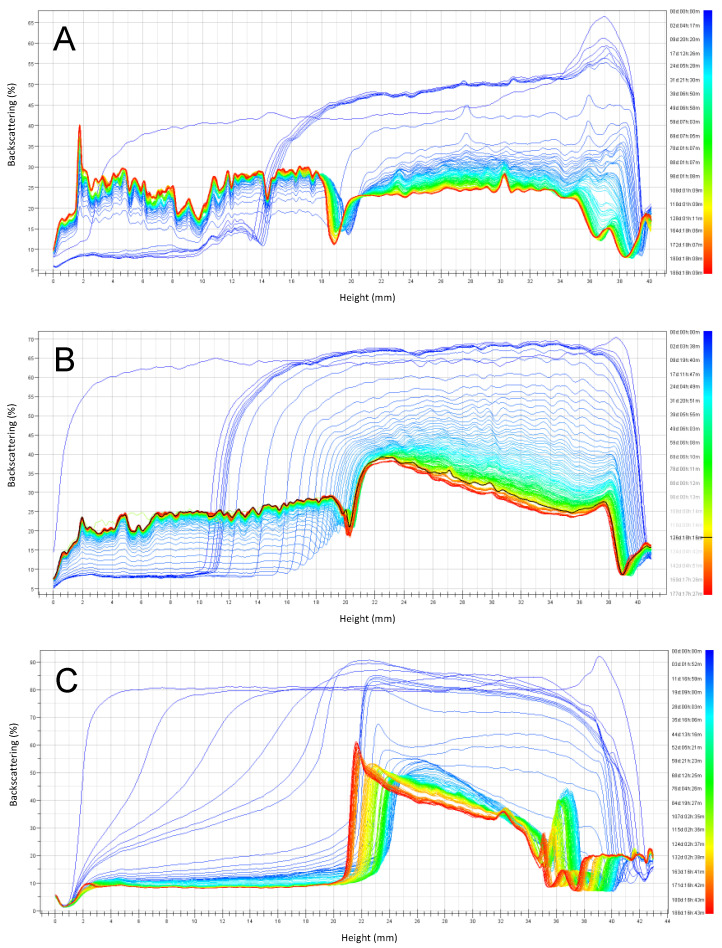
Emulsions backscattering profiles prepared at pH 3 and stored six months at 30 °C. (**A**) Emulsions stabilized by rapeseed protein precipitated at pH 3 (RPP3), (**B**) Emulsions stabilized by rapeseed protein precipitated at pH 6.5 (RPP6.5), (**C**) Emulsions stabilized by soy lecithin.

**Figure 9 foods-10-01657-f009:**
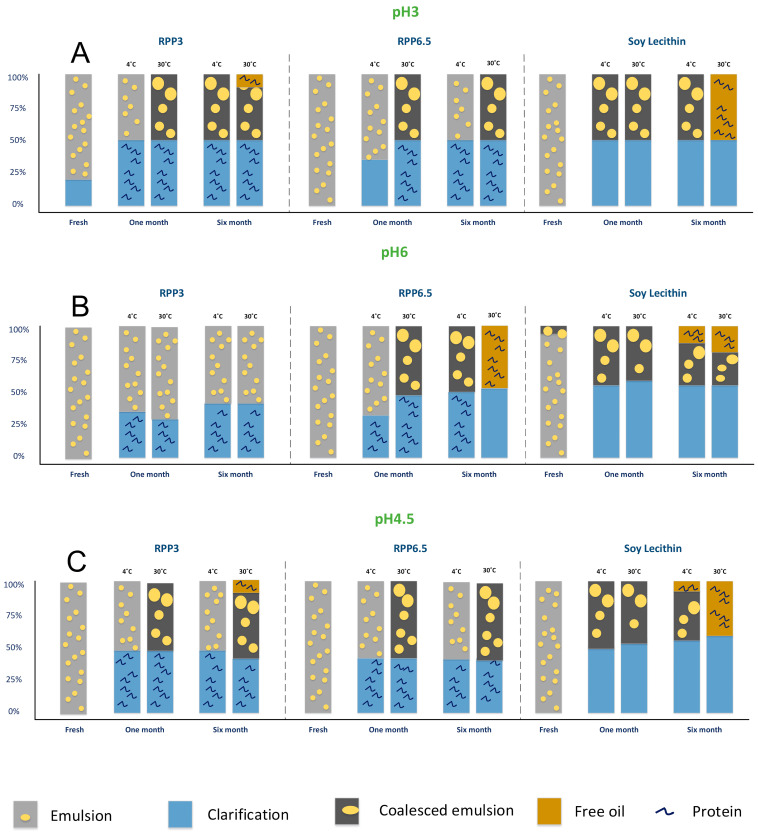
Schematic diagram of the phases observed for emulsions prepared with RPP3, RPP6.5 and soy lecithin after six months of storage at 4 and 30 °C. (**A**) Emulsion pH 3, (**B**) Emulsion pH 6, (**C**) Emulsion pH 4.5.

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
