# Peer review of "The Effect of pH and Storage Temperature on the Stability of Emulsions Stabilized by Rapeseed Proteins"

_foods, 2021, doi:10.3390/foods10071657_

Round 1

Reviewer 1 Report

In the section „Material and methods (statistical analysis) authors did not indicated whether and with which test(s) the equality of variance was checked.

The statistical tests performed were a one-way ANOVA rather than an univariate general model. Correct. L219-240. No reference to statistical analyzes in the text.

Figure 1. Start marking the highest values with zeta potential with a and marking the next values with consecutive letters of the alphabet. Were all values compared with each other? Letters in the Figure 1 indicate this, but presumably only the effect of pH was compared. There is no x axis label/scale.

L247. If the droplet size was significantly different  p<0.05, not p>0.05. Figure 2. as in Figure 1.

Author Response

Reviewer number 1

Thank you for the valuable comments. We have done our best to make the suggested changes and hope you find them satisfactory in the revised manuscript. Please find our responses to the comments below. In the revised manuscript new and significantly modified sections are indicated in yellow.

Comment #1: In the section „Material and methods (statistical analysis) authors did not indicated whether and with which test(s) the equality of variance was checked.

Response: The equality of variance was checked with the Shapiro-Wilks test and the information has been added at line 205-206.

Comment #2: The statistical tests performed were a one-way ANOVA rather than an univariate general model. Correct. L219-240. No reference to statistical analyzes in the text.

Response: Thanks for pointing this out, the statistical test we used is called General Linear Model (GLM), and it has been corrected at line 208. Statistical analysis has been referred to at line 230, 231, 254, 263, 287.

Comment #3: Figure 1. Start marking the highest values with zeta potential with a and marking the next values with consecutive letters of the alphabet. Were all values compared with each other? Letters in the Figure 1 indicate this, but presumably only the effect of pH was compared. There is no x axis label/scale.

Response: We agree of the disposition of the lower case letters, and have reported the highest zeta potential with a and so on. See the revised Fig 2. The figure number is changed due to a suggestion from the other reviewer to add a graphical illustration of the study and this new figure is figure 1. All values were compared with each others. The x-axis was there but was kind of hidden in the middle. We have put the x legend at the bottom of the graph instead to increase clarity.   

Comment #4: L247. If the droplet size was significantly different  p<0.05, not p>0.05. Figure 2. as in Figure 1.

Response: The correct sign has been added at line 252. Thanks for pointing this out.

Reviewer 2 Report

The present paper explained the rapeseed proteins isolated from Rapeseed press cake (RPC), and their emulsions for food application. The authors thoroughly studied the necessary parameters. 

  1. Authors should give graphical or diagrammatic representations of the process, it will better for the readers to understand the concept more clearly.
  2. Figure 1-2 is not clear, what is the scale of the x-axis, its better if authors can draw the line for x-axis scales (dotted line is fine). What are the points a, b,.....f.  Indicate clearly in Fig, legend. Also, increase the size of the Fig. Make it large.
  3. The authors should comprehensively compare the other protein-based emulsions for this study. 
  4. In the conclusion section, the author should focus on how this can be applicable and technical constraints if any..

Author Response

Reviewer number 2

Thank you for the valuable comments. We have done our best to make the suggested changes and hope you find them satisfactory in the revised manuscript. Please find our responses to the comments below. In the revised manuscript new and significantly modified sections are indicated in yellow.

Comment #1: Authors should give graphical or diagrammatic representations of the process, it will better for the readers to understand the concept more clearly.

Response: Thanks for the good idea, we have added a graphical representation of the process of emulsions formulation and characterization (Fig 1).

Comment #2: Figure 1-2 is not clear, what is the scale of the x-axis, its better if authors can draw the line for x-axis scales (dotted line is fine). What are the points a, b,.....f.  Indicate clearly in Fig, legend. Also, increase the size of the Fig. Make it large.

Response: The x-axis was hidden in the middle of the Fig 1. We have moved it to the bottom to increase clarity. A dotted line has been added to visualize zero mV. The lowercase letters symbolize the significant difference between the data points has been clarified in the figure legends. Also, the size of the figures has been increased. 

Comment #3: The authors should comprehensively compare the other protein-based emulsions for this study. 

Response: Results from other research works where proteins were used as stabilizers alone or in combination with other stabilizing compounds such as surfactants or polysaccharides are compared with the work made in the present study in terms of droplet size, droplet size distribution and stability. The new sections can be found at lines 317-327 and 459-467.

The references included in the revised version are the following:

D.G. Dalgleish, 1997, Adsorption of protein and the stability of emulsions, Trends in Food Science and Technology, 8, 1-6

  1. Tong, J. Cao, M. Sun, P. Liao, S. Dai, W. Cui, X. Cheng, Y. Li, L. Jiang, H. Wang, (2021) Physical and oxidative stability of oil-in-water (O/W) emulsions in the presence of protein (peptide): Characteristics analysis and bioinformatics prediction, LWT - Food Science and Technology 149 a111782

l.C.B., Züge, C. . I. H., G.M. Maciel, J. L. M. Silveira, A. P. Scheer, (2013) Catastrophic inversion and rheological behavior in soy lecithin and Tween 80 based food emulsions, Journal of Food Engineering 116, 72–77

  1. Liu, R. Pei, L. Peltonen, M. Heinonen, (2020), Assembling of the interfacial layer affects the physical and oxidative stability of faba bean protein-stabilized oil-in-water emulsions with chitosan, Food Hydrocolloids 102, a105614
  2. Zhu, Q. Xu, X. Liu, Y. Xu, L. Yang, S. Wang, J. Li, Y. He, H. Liu, (2020), Soy glycinin-soyasaponin mixtures at oil–water interface: Interfacial behavior and O/W emulsion stability, Food Chemistry,327, 127062
  3. Burgos-Díaz, T. Wanderslebe, A. M. Marqués, M. Rubilar, (2016) Multilayer emulsions stabilized by vegetable proteins and polysaccharides, Current Opinion in Colloid & Interface Science 25, 51–57

D.J. McClements, (2007), Critical Reviews in Food Science and Nutrition, Critical Reviews in Food Science and Nutrition, 47:611–649 (2007)

Comment #4: In the conclusion section, the author should focus on how this can be applicable and technical constraints if any.

Response: Applications and challenges have been added to the conclusion section, see line 484-487.  
